# A Visible-Light-Active CuS/MoS_2_/Bi_2_WO_6_ Aptamer Sensitively Detects the Non-Steroidal Anti-Inflammatory Drug Diclofenac

**DOI:** 10.3390/nano12162834

**Published:** 2022-08-18

**Authors:** Yun He, Hongjie Gao, Jiankang Liu

**Affiliations:** The Key Laboratory of Biomedical Information Engineering of Ministry of Education, School of Life Science and Technology, Xi’an Jiaotong University, Xi’an 710049, China

**Keywords:** photoelectrochemical, CuS/MoS_2_/Bi_2_WO_6_, diclofenac (DFC), heterostructure, aptasensors

## Abstract

Diclofenac is a non-steroidal, anti-inflammatory drug and is clinically used for the treatment of osteoarthritis, non-articular rheumatism, etc. This research aimed to demonstrate the creation of an upgraded photoelectrochemical (PEC) aptamer sensor for detecting diclofenac (DCF) with high sensitivity. In this work, photoactive materials and bio-identification components served as visible-light-active CuS/MoS_2_/Bi_2_WO_6_ heterostructures and aptamers, respectively. CuS and MoS_2_/Bi_2_WO_6_ were combined to improve photocurrent responsiveness, which helped the structure of PEC aptasensors. Additionally, the one-pot synthesis of CuS/MoS_2_/Bi_2_WO_6_ was ecologically beneficial. With these optimizations, the photocurrent response of aptamer/CS/CuS/MoS_2_/Bi_2_WO_6_ exhibited linearity between 0.1 and 500 nM DCF. The detection limit was 0.03 nM (S/N = 3). These results suggest that the PEC sensing technique might produce an ultra-sensitive sensor with high selectivity and stability for DCF detection.

## 1. Introduction

Diclofenac (DCF) is a non-steroidal, anti-inflammatory drug (NSAID) marketed for global consumption [1]. It has been utilized for surgical procedures such as gynecological, orthopedic, and ontological procedures [2]. Additionally, DCF specializes in the treatment of osteoarthritis, renal and biliary cramps, pharyngotonsillitis, otitis, annexites, and primary dysmenorrhea, as well as various infection-related pain operations [1,2,3]. According to a study in 2018, around half of the original medication was metabolized; moreover, 70% of pharmaceuticals was excreted via urine, despite diverse conditions occurring in relation to those targeted drugs [4]. The risk assessment of drug toxicity revealed that, according to data on the DCF presence in wastewater following standard biological treatment, around 80% of the original pollutant was retained [5]. Inefficient effluent treatment plants that failed to eliminate drugs from wastewater were utilized to develop four drug detection techniques.

Many methods have been employed for DCF detection, such as immunoassays, chromatographic techniques, and electrochemical methods [6,7,8]. Clearly, these techniques, which provide the efficient detection of DCF at concentrations ranging from M to nM, are not without limitations, including the requirement for costly equipment. Additionally, they consume significantly more time and need complicated preprocessing, limiting their practical applicability [9,10]. As a result, a sensitive and selective sensor capable of providing more promptness and ease for DCF detection needs to be built.

Photoelectrochemical (PEC) sensors, which specialize in high sensitivity, simply structured instruments, easily conducted downsizing, and low prices, have previously attracted academics to conduct research on them [11,12,13]. When developing PEC sensors, improved photocatalyst performance contributes to increased sensitivity. Nonetheless, PEC sensing must rely on photocatalytic processes that lack analyte specificity. To overcome the issue, significant efforts have been made to improve the selectivity of PEC sensors by including numerous recognition components such as molecular-imprinted polymers, enzymes, antibodies, and aptamers [14,15]. In terms of these factors, aptamers, well-known antibody mimetics with excellent recognition capacity over particular targets, have been used to build highly selective PEC aptasensors orientated with a more extensive variety of analytes consisting of inorganic ions, proteins, cells, antibiotics, and chemical substances.

Some bismuth-based semiconductors also have a hybrid orbital of Bi 6s and O 2p, decreasing the bandgap while generating a deep valance band, such as BiOI, Bi_2_S_3_, Bi_2_XO_6_ (X = Mo, W), and others [16,17]. These elements, characterized by abundance, low toxicity, and low cost, have become the target of intense studies in PEC research [18]. Furthermore, the morphological structure of Bi_2_WO_6_, which includes Bi_2_O_2_ layers and WO_6_ octahedral structure layers, has a significant impact on its physical and chemical characteristics [19]. Furthermore, Bi_2_WO_6_ is one of the possible visible-light-driven photocatalysts with a moderate band gap (approximately 2.5 eV) [20]. More Bi_2_WO_6_ uses emerged in recent years, including PEC solar cells, PEC sensors, and photoelectrocatalytic hydrogen generation [21,22,23]. However, high-degree recombination specialized in photoexcited carriers, comparably weak light consumption capacity in pure Bi_2_WO_6_, and specialized poor photocatalytic activity severely restricted the highlighted practical uses in the environment [24]. As a result, significant efforts have been undertaken to address these shortcomings. Semiconductor coupling and metal particle deposition coupling with carbon materials are two examples [23,24].

MoS_2_ exhibits a photocatalytic capability due to its differentially specialized structure of sandwich layers of S–Mo–S atoms [25]. Furthermore, according to the current study, MoS_2_ has better photocatalytic activity because to its differentiated electrical and optical characteristics and large surface area [26]. As a result, MoS_2_ was used in numerous materials, including BiVO_4_ [18], WS_2_ [26], and TiO_2_ [27], to increase electron–hole pair separation and PEC performance. Semiconductor chalcogenides have attracted tremendous attention with high abundance and low cost. Copper sulfide (CuS), which performs the duty of p-type semiconductors, presents availability, versatility, low toxicity, etc. [28]. Moreover, it also has unique electronic and optical properties, with additional applications to catalysis, solar cell, sensing, and lithium-ion batteries. CuS, a narrow-band gap semiconducting material (approximately 1.7 eV), is employed for coupling with bismuth-based semiconductors for enhanced photocatalytic performance [29,30].

Based on the abovementioned background research, a CuS/MoS_2_/Bi_2_WO_6_ photoactive material was developed. By demonstrating greater photocurrents, CuS/MoS_2_/Bi_2_WO_6_ extended the light response to the visible region beyond the elements of Bi_2_WO_6_ and MoS_2_/Bi_2_WO_6_ alone. As a result of the increased photo-to-current efficiency, it was proven that Bi_2_WO_6_ sensitized with CuS and MoS_2_ may promote photogenerated electron-hole pair separation. The improved CuS/MoS_2_/Bi_2_WO_6_ photoactive material enabled the unification of a large linear range with a low detection limit, resulting in a successful PEC aptasensor for DCF detection. This novel PEC sensing strategy provides a highly selective and robust detection method for DCF detection. Moreover, the sensor also has good application prospects in practical applications. This also demonstrates the enormous potential of CuS/MoS_2_/Bi_2_WO_6_ nanocomposites in fields such as PEC sensing, photocatalysis, and others.

## 2. Experimental Section

### 2.1. Materials and Reagents

Thiourea (SC(NH_2_)_2_), ethanol (99.7%), bismuth nitrate pentahydrate (Bi(NO_3_)_3_·5H_2_O), copper nitrate trihydrate (Cu(NO_3_)_2_·3H_2_O), sodium tungstate dihydrate (Na_2_WO_4_·5H_2_O), glacial acetic acid (99.7%), glutaraldehyde (GA, 50%), and tris (hydroxymethyl) aminomethane (Tris) were purchased from Sinopharm (Shanghai, China). Chitosan (CS, 95%) was obtained from Sigma-Aldrich (America). DCF sodium drugs were obtained from Aladdin (Beijing, China). All other reagents were of analytical reagent grade. The amino-functionalized DCF aptamer (5′-NH_2_-TCTA ACGT GAAT GATA GACC TGGC TTGG GTGG TGGG CGAC TGAC TGGC GGTG CAAC GTTA ACTT ATTC GACC ATA-3′) was composed by the Shanghai Sangon Biotech Co., Ltd. (Shanghai, China) and purified via an HPLC technique. The aptamer solution was compounded by dissolving aptamer into the Tris-HCl buffer (0.1 M, pH 7.4). Phosphate-buffered solution (PBS, 0.1 M, pH 7.4) was prepared from Na_2_HPO_4_ and NaH_2_PO_4_ and exploited as an electrolyte during the period of detection, while double-distilled water (Milli-Q, Millipore) was used for all aqueous solutions throughout the experiment. The rpm of the washing steps was 8000 r/min.

### 2.2. Preparations of All Materials

Bi_2_WO_6_ material synthesis: 2 mmol of Bi(NO_3_)_3_.5H_2_O (0.9702 g) was added to ultrapure water (40 mL). Then, 1 mmol of Na_2_WO_4_·5H_2_O (0.3838 g) was added to obtain the suspension slowly and appropriately under continuous stirring. Subsequently, the mixed suspension received another 1 h of stirring. After being transferred to a Teflon-lined autoclave (100 mL), the mixed solution underwent 20 h of heating treatment at 180 °C. At last, deionized water together with ethanol helped to wash the product three times, followed by drying treatment at 60 °C.

MoS_2_/Bi_2_WO_6_ composite synthesis: 1mmol of Bi_2_WO_6_ was added to 20 mL of ultrapure water, and different contents of (NH_4_)_6_Mo_7_O_24_∙4H_2_O (0.1412 mg, 3.531 mg, and 5.2965 mg) and SC(NH_2_)_2_ (0.1218 mg, 0.3044 mg, and 0.4568 mg) were added to the suspension obtained above with continuous stirring. After being transferred to a 50 mL Teflon-lined autoclave, the mixed solution underwent 24 h of heating at 200 °C. The obtained samples were washed and calcined. The material was named MoS_2_/Bi_2_WO_6_-2%, MoS_2_/Bi_2_WO_6_-5%, and MoS_2_/Bi_2_WO_6_-7% (2%, 5%, and 7% = m(MoS_2_)/[m(MoS_2_) + m(Bi_2_WO_6_)]).

CuS/MoS_2_/Bi_2_WO_6_ composite synthesis: 1 mmol of MoS_2_/Bi_2_WO_6_ (0.6711 g) was added to 20 mL of ultrapure water, and different contents of SC(NH_2_)_2_ (4.5672 mg) and Cu(NO_3_)_2_·3H_2_O (14.496 mg) were added to the suspension obtained above, which were stirred continuously. Subsequently, the mixed suspension received another 30 min of stirring, and then, 10 mL of ethanol solution was added. After being transferred to a 50 mL Teflon-lined autoclave, the mixed solution underwent 12 h of heating at 200 °C.

### 2.3. Characterization

A Bruker D8 Advance diffractometer (Billerica, MA, USA) equipped with Cu Kα radiation was applied to perform the X-ray diffraction (XRD) (λ = 0.154056 nm). An ESCALAB 250Xi (Thermo Fisher Scientific, Waltham, MA, USA) was employed to carry out the X-ray photoelectron spectroscopy (XPS). A UV–visible (UV–vis) spectrophotometer was employed to collect the diffuse reflection spectra (DRS) exhibited by these materials with BaSO_4_ as the background between 200 and 800 nm. Scanning electron microscopy (SEM) together with transmission electron microscopy (TEM, at 200 kV) were used to characterize the morphology exhibited by these samples.

### 2.4. Construction of PEC Aptasensor

Indium tin oxide (ITO, 1 cm × 2 cm) electrodes were ultrasonically cleaned sequentially in NaOH solution (0.1 M), ethanol, and ultrapure water. They were then dried under infrared light. The PEC aptasensor was prepared using the following steps. Firstly, 20 μL of CuS/MoS_2_/Bi_2_WO_6_ suspension (6 g/L, 500 mL) was modified onto ITO within a fixed geometric region (0.5 cm × 1 cm) and dried under room temperature to form a CuS/MoS_2_/Bi_2_WO_6_ electrode. Then, 20 μL of CS (0.05%), acting as fixing agent, was dropped onto the CuS/MoS_2_/Bi_2_WO_6_ and dried off at room temperature. Next, 10 μL of 2.5% GA aqueous solution (a cross linker for the amino functional aptamer) was put onto the above electrode and kept for 1 h at 25 °C, followed by rinsing with PBS (0.1 M, pH 7.4) to eliminate excess GA. The resultant surface was coated with 10 μL of amine-functionalized aptamer solution (1 μM) and incubated at 4 °C for 12 h. The prepared sensor of aptamer/CS/CuS/MoS_2_/Bi_2_WO_6_ was washed with PBS to eliminate any unbounded aptamer. It should be highlighted that the NH_2_ group in the DCF aptamer was covalently attached to the NH_2_ group of the immobilized CS on the ITO surface, using GA as the linking agent. Ultimately, the PEC aptasensors were obtained.

### 2.5. Electrochemical Experiments

The measurements of the PEC were made with a CHI 660E electrochemical workstation (CH Instrument Company, Shanghai, China). A conventional three-electrode system cell was employed. Pt wire was used as the counter electrode, a saturated calomel electrode (SCE) was the reference electrode, and ITO glass was the working electrode. The light source came from a xenon lamp (PLS-SXE 300, 100 mW cm^−2^, *λ* ≥ 420 nm), and the light source was kept at 15 cm while the modified electrode was applied in the PEC system to detect DCF at an operating potential of 0.1 V. Electrochemical impedance spectroscopy (EIS) was performed in PBS (0.1 M, pH = 7). To investigate the detection performance of the PEC aptasensor, the prepared aptamer/CS/CuS/MoS_2_/Bi_2_WO_6_ electrode was incubated with 20 μL of DCF solution at various concentrations for 40 min.

### 2.6. Computational Methodology

First-principle calculations were carried out using density functional theory (DFT) with generalized gradient approximation (GGA) of Perdew–Burke–Ernzerhof (PBE) implemented in the Vienna Ab-Initio Simulation Package (VASP) (Figure 1). The valence electronic states were expanded on the basis of plane waves with the core–valence interaction represented using the projector augmented plane wave (PAW) approach and a cutoff of 520 eV. A Γ-centered k-mesh of 3 × 3 × 1 points was used for the calculations. Convergence was achieved when the forces acting on ions became smaller than 0.02 eV/Å.

## 3. Results and Discussion

### 3.1. Physical Characterization

The crystal and phase information of synthetic materials were determined using X-ray diffraction (XRD) (Figure 1 and Appendix A). Broad peaks revealed the CuS XRD pattern. The typical diffraction peaks at 27.52, 31.78, and 53.04° were due to the crystal faces of (100), (103), and (108) interlayer reflections. MoS_2_/Bi_2_WO_6_ diffraction peaks were found on the orthorhombic phase of Bi_2_WO_6_ (JCPDS No. 39-0256) and the hexagonal phase of MoS_2_ (JCPDS No. 37-1492). However, typical CuS diffraction peaks did not appear in the CuS/MoS_2_/Bi_2_WO_6_ heterostructure. This may have been due to the low content of CuS in CuS/MoS_2_/Bi_2_WO_6_.

XPS (Figure 2) confirmed the existence of Cu, S, O, Bi, W, and Mo in the chemical composition and valence band structure of CuS/MoS_2_/Bi_2_WO_6_-5%. The high-resolution spectrum of Bi (Figure 2A) exhibited two distinct peaks for electrons in the Bi 4f orbitals at 158.8 and 164.2 eV under the distributions on Bi 4f7/2 and Bi 4f5/2 [31]. Figure 2B depicts the spectra for O 1 s which is resolved into peak positions captured at 529.2 and 530.9 eV corresponding to the lattice oxygen in the microstructure and surface-adsorbed oxygen from the atmosphere, respectively [32]. The distinctive peaks at 34.9 eV and 36.9 eV for W 4f7/2 and W 4f5/2 (Figure 2C) were due to the presence of W atoms in the +6 oxidation state [33]. Figure 2D matches Mo 3d3/2 at 231.6 eV and Mo 3d5/2 at 228.4 eV with Mo^4+^ [34]. The XPS spectra of Cu 2p is shown in Figure 2E, with peaks centered at 932.4 eV (Cu 2p3/2) and 952.4 eV (Cu 2p1/2), indicating that Cu is present in the Cu^2+^ state [35,36]. According to Figure 2F, the doublet peak at 167.9 eV was a candidate for S 2p1/2, while the peak at 162.5 eV was for S 2p3/2, indicating the presence of metal sulfides [37,38]. These XPS data and Appendix A were consistent with the composition CuS/MoS_2_/Bi_2_WO_6_-5%.

Figure 3 represents the TEM images of their nanocomposite, and Figure 3A,B depict the recording of the Bi_2_WO_6_ and CuS/MoS_2_/Bi_2_WO_6_-5% microstructure and high-resolution imaging. The picture demonstrates the microstructure’s density, as indicated by the differential flakes protruding from the floral core structure. Additionally, to illuminate the growth planes in detail, high-resolution TEM (HR-TEM) in Figure 3D was used to calculate the interplanar spacing for 0.316 nm following the (131) plane of the Bi_2_WO_6_ structure, due to the lattice spacing of 0.328 and 0.158 for overgrown CuS and MoS_2_, respectively, which arose from the (100) and (110) planes. Among Bi_2_WO_6_, CuS, and MoS_2_, an explicit interface that benefited photocatalysis by forming a heterojunction enabled the efficient transfer of charge carriers.

UV–visible diffuse reflectance spectroscopy was used to characterize the optical attributes of Bi_2_WO_6_, MoS_2_/Bi2WO_6_-5%, and CuS/MoS_2_/Bi_2_WO_6_-5% (Figure 4 and Appendix A). According to Figure 4A, after MoS_2_ and CuS were included in the structure of Bi_2_WO_6_, the visible light range offered a location for the broad absorption band. This might have been due to the enhanced visible-light absorptivity conferred by MoS_2_ and CuS [39]. Accordingly, CuS- and MoS_2_-sensitized Bi_2_WO_6_ was designed to exhibit superior photocatalytic activity in the visible area due to heterojunction formation, sensitivity, and rapid charge transfer kinetics [40]. The room temperature photoluminescence emission spectra were measured in order to investigate the charge separation properties of Bi_2_WO_6_, MoS_2_/Bi_2_WO_6_-5%, and CuS/MoS_2_/Bi_2_WO_6_-5%. Figure 4B depicts the photoluminescence emission spectrum at a 350 nm excitation wavelength. Bi_2_WO_6_ was seen to have a higher characteristic spectrum between the wavelengths of 500 and 680 nm, which was caused by charge carrier recombination, the emission peak position in the range of 550–600 nm shift to the right when adding CuS to the composite MoS_2_/Bi_2_WO_6_, which is due to the smaller energy band of the composite material, and the red shift of the emission peak occurred [41]. Obviously, the peak intensity of MoS_2_/Bi_2_WO_6_-5%, and CuS/MoS_2_/Bi_2_WO_6_-5% was significantly lower than that of Bi_2_WO_6_, suggesting that the integration of CuS and MoS_2_ could reduce photogenerated charge carrier recombination. In general, reduced intensity indicates improved electron–hole separation efficiency due to the carriers’ greater lifespan and higher PEC.

### 3.2. Theoretical Calculations

DFT determined the density of states (DOS) and energy band structures for Bi_2_WO_6_ and CuS/MoS_2_/Bi_2_WO_6_-5% (Figure 5). Clearly, Bi_2_WO_6_ serves as the photocatalytic material of an indirect band-gap semiconductor since the valence band maximum (VBM) and conduction band minimum (CBM) are located at diverse high symmetry points, as seen between CBM at Г-point and VBM at X-point. Calculations yielded a Bi_2_WO_6_ band gap of 1.86 eV. Notably, the band gaps were significantly lower compared with the literature (2.56 eV for Bi_2_WO_6_) [35], which was likely due to the defected GGA function (Figure 5A,B). In the photocatalyst, photogenerated h+ and e- were efficiently isolated and quickly transported onto surfaces. Figure 5B,D shows the DOS with CuS/MoS_2_/Bi_2_WO_6_-5%, which indicated that CuS and MoS_2_ gain charges as electron trapping and shuttling sites, suppress the recombination of electrons/holes, and promote electron separation and transfer [36].

### 3.3. Photoelectrochemical Sensor

Electrochemical impedance spectroscopy (EIS) was used to analyze the electrode contact properties. The impedance spectra following the various biosensor fabrication procedures are shown in Figure 6A. Every impedance spectrum contained a high-frequency semicircle and a low-frequency linear part. The semicircle represented a finite electron transfer mechanism, whereas the linear section represented a finite diffusion process. The semicircle’s diameter illustrated the restricted diffusion of the redox probe into the electrode interface, with the exact quantities as the electron transfer resistance (Ret). The impedance spectra of the MoS_2_/Bi_2_WO_6_ electrode represented a tiny semicircle (Appendix A), which corresponded to a modest Ret value. Because of the poor conductivity of this semiconductor, the Ret (curve I) for CuS/MoS_2_/Bi_2_WO_6_-5% increased. As a result of the decreased electron transport efficiency, the Ret saw extensive growth when CS was added (curve II). During the aptamer fixing, the Ret increased further (curves III), owing to the poor conductivity of these organic molecules and resistance from the negatively charged layer of phosphate groups. Then, after incubating the electrode with DCF, the Ret increased dramatically (curve IV), indicating DCF contact with the electrode surface. The biosensor was successfully created due to the potential shift in Ret.

The low photocurrent intensity shown by the CuS/MoS_2_/Bi_2_WO_6_-5% electrode resulted from a high electron–hole pair recombination rate (Figure 6B). Because of the expected acceleration of electron transport, the photocurrent intensity increased significantly after adding CuS nanoparticles (curve II). When the electron donor solution of ascorbic acid (AA) and the electrode surface of CuS/MoS_2_/Bi_2_WO_6_-5% met with the impeded electron exchange, the photocurrent density gradually decreased when CS and aptamer (curves III and IV) were added. The aptamer sensor was found be successfully constructed. The binding of DCF resulted in increased photocurrent (curve IV), which most likely resulted in DCF oxidation and electron transport to the counter electrode.

When the aptamer recognizes the detection target, the magnitude of change in the photocurrent plays a critical role in the sensitivity of the photoelectrochemical aptamer analysis. This study shows the aptamer analysis method’s design. The photocurrent response characterization experiments described above specialized in ultra-high sensitivity, which stems from the two elements listed below. The schematic diagram could be used to represent the preparation process of the intended aptamer sensor in order to obtain an explicit interpretation (Figure 2) [42,43,44]. In the absence of DCF, the electron transfer rate may have been considerably enhanced because the CuS/MoS_2_/Bi_2_WO_6_-5% composite could absorb the energy of the UV and light source. As an electron donor, ascorbic acid could consume holes in the semiconductor material, reduce the electron–hole recombination rate of the material, and could amplify the photocurrent signal, thereby reducing the error caused by the small change in the current data after adding DCF. Furthermore, the sensing electrode altered the recombination of electron–hole pairs and showed a noticeable photocurrent response. Further oxidization was carried out under the photogenerated holes when the aptamer connected with the target DCF. As a result, the sensing electrode’s current was accelerated, resulting in a significant increase in photocurrent intensity.

### 3.4. The Influence of Effective Parameters on the Detection of DCF

Figure 7A depicted the examination of the applied potential selection. Due to the difference in potential between −0.2 V and 0.3 V, the aptamer/CS/CuS/MoS_2_/Bi_2_WO_6_-5% and DCF/aptamer/CS/CuS/MoS_2_/Bi_2_WO_6_-5% were tested in the dark or in the presence of light. When comparing photocurrent to dark in different currents, the photocurrent decreased dramatically when the potential changed from −0.2 V to 0.1 V. However, it was shown to exhibit a modest drop when the potential difference was between 0.1 V and 0.3 V. As a result, 0.1 V may be used as the PEC matching voltage.

During the detection process, the pH of the electrolyte, a critical component of the PEC performance, must be optimized (Figure 7B). The photocurrent increased as the pH value of the electrolyte was increased from 5 to 7. As the pH value decreased from 7 to 9, the photocurrent decreased. Because the neutral environment presumably benefits the aptamer activity, it could achieve its maximum value at pH = 7.

According to Figure 7C, to conduct in-depth research on the long-term stability of DCF detection based on the aptamer/CS/CuS/MoS_2_/Bi_2_WO_6_-5% sensor in the absence and presence of DCF, in the same solution, a sensor coated with aptamer/CS/CuS/MoS_2_/Bi_2_WO_6_-5% was used to measure DFC every 5 days. After 20 days, there was no substantial change in the aptamer/CS/CuS/MoS_2_/Bi_2_WO_6_-5% sensor’s photocurrent response to DCF. As a result, it was demonstrated that the composite sensor was designed to emphasize improved stability. To achieve optimal sensitivity, the aptamer concentration (0.1–2 µmol/L) is optimized in Figure 7D, which demonstrates the electrode’s greatest photocurrent at a concentration of 1.0 µmol/L. As a result, this study approved a 1.0 µmol/L aptamer concentration.

Aptamer/CS/CuS/MoS_2_/Bi_2_WO_6_-5% was used to analyze DCF concentrations under optimized circumstances. According to Figure 8A, the photocurrent increased in proportion to the DCF concentration due to the aptamer’s specific binding to DCF. The generated DCF–aptamer complexes on the sensor interface reduced the steric barrier for electron-donor diffusion, resulting in a drop in photocurrent. Within the concentration range of 0.1 to 500 nM (Figure 8B), the photocurrent change followed a linear relationship with the DCF concentration, resulting in a detection limit of 0.03 nM (S/N = 3). PI = 1.278 + 1.316logC was used as the calibration regression equation (nM). A correlation coefficient (*R*^2^) of 0.9965 was obtained. The suggested sensor’s performance, which included additional previously reported characteristics, demonstrated a substantially lower detection limit within a comparable linear range (Appendix A).

According to Figure 8C, after waiting for the photocurrent of aptamer/CS/CuS/MoS_2_/Bi_2_WO_6_-5% to stabilize, additional interfering agents, including kanamycin (KAN), sulfadimethoxine (SDM), ochratoxin A (OTA), dopamine (DOP), glucose (GLU), Al^3+^, and Fe^2+^, were added, and the change in the photocurrent after stabilization was investigated. The photocurrent showed no discernible change in the coexisting ion supplement. Additionally, it demonstrated high selectivity for DCF detection using aptamer/CS/CuS/MoS_2_/Bi_2_WO_6_-5%. Figure 8D demonstrates the stability of the aptamer/CS/CuS/MoS_2_/Bi_2_WO_6_-5%-based PEC sensor for DCF detection. The repeated photoexcitation procedure monitored the PEC sensor every 800 seconds. The observation demonstrated that the photocurrent and dark current remained stable throughout the time, indicating the reliability of the PEC sensor during the detection of DCF. As discussed before, the aptamer/CS/CuS/MoS_2_/Bi_2_WO_6_-5%-based sensor has exceptional stability, repeatability, and DCF stability.

### 3.5. Real Sample Analysis

By using a standard addition technique and comparing it to HPLC readings, DCF in tap water was analyzed to determine the practical applicability of the created biosensor. To begin, DCF capsules were pulverized to a powder in an agate mortar. Following that, different volumes of powder were dissolved in tap water to create solutions with varying concentrations. Then, on a quantitative basis, the resulting solutions were analyzed, as shown in Table 1. Consistent with the HPLC recovery (95.00–101.82%), the aptasensor recovery was between 97.00 and 102.33%, demonstrating that the PEC aptasensor can be utilized to detect DCF in realistic samples.

## 4. Conclusions

To summarize, our study developed a new visible-light PEC aptamer sensor based on CuS/MoS_2_/Bi_2_WO_6_ nanocomposites for DCF detection. DCF is clinically used for osteoarthritis and non-articular rheumatism, in which the PEC aptamer sensor will have great application prospects. The CS/CuS/MoS_2_/Bi_2_WO_6_ heterostructure plays a critical role in the photocurrent response of the PEC sensor, which operated across a broad linear range of 0.1–500 nM with a detection limit of 0.03 nM, exceeding the detection limit and linearity of typically modified electrodes. This novel PEC sensing strategy provided an ultra-sensitive sensor with high selectivity and stability for DCF detection. At the moment, our focus is on miniaturizing this technology to be used in more areas.

## Data Availability

The data presented in this study are available on request from the corresponding author.

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
