# Peer review of "A Visible-Light-Active CuS/MoS2/Bi2WO6 Aptamer Sensitively Detects the Non-Steroidal Anti-Inflammatory Drug Diclofenac"

_nanomaterials, 2022, doi:10.3390/nano12162834_

Round 1

Reviewer 1 Report

The manuscript reports on a photoelectrochemical sensor based on aptamer-inorganic composite. In the paper, the preparation of main components and construction of this aptasensor for detecting diclofenac is claimed. The sensor obtained was studied for the photoelectrochemical response at different working conditions (applied potential, pH and ion supplement). The paper is within the scope of Nanomaterials journal, its results are of certain interest for the readership of the journal. The manuscript is suitable for publication in Nanomaterials journal upon the revision.

  1. In the experimental section, the synthesis conditions (temperature and duration) for preparation of Bi2WO6, MoS2/Bi2WO6 and CuS/MoS2/Bi2WO6 are different (20 or 24 h, 180 or 200C). Please specify why such conditions of synthesis at each stage?
  2. How was the CS/CuS/MoS2/Bi2WO6 suspension obtained? The first step of construction of PEC aptasensor isn't quite clear.
  3. Please specify the chemical composition of anatase polymorph, is it referred to CuS, MoS2 or Bi2WO6? Anatase is TiO2, no titania in the material.
  4. The JCPDS card №39-0256 relates to the orthorhombic phase of Bi2WO6, not tetragonal. The JCPDS card №37-1492 state the hexagonal phase of MoS2, not tetragonal.
  5. From the experimental part, it is unclear which component is 5%. Are these mass or molar percentages? The Supplementary also contain the data about 2 and 7% of the component. Why was the composition containing exactly 5% chosen as the main material?
  6. The chemical composition of the composite (CS/CuS/MoS2/Bi2WO6) should be established. FTIR spectrum is necessary.
  7. In the XPS results, some energy values in the text differ from the energy values on the XPS spectra. Please specify if this is experimental or literature data.
  8. The paper describes satellite peak for copper at 941.5 eV, but it is not observed on Figure 2E. Please correct.
  9. The text shows the values of the interplanar spacing for (002) of MoS2, and (110) is highlighted on the HRTEM image (Figure 3D). It should be noticed that the interplanar spacing for (110) plane of MoS2 is 0.158 nm, not 0.615 nm (this is true for the (002) plane), as it shown on Figure 3D. Please mark the plane (110) on the HRTEM image or change the text.
  10. Please bring the scale labels on the SEM and TEM images to a uniform view.
  11. Please complete the caption to Figure 3: HRTEM image ... (D).
  12. Please estimate the value of the band gap of the obtained materials from the diffuse reflection spectra. How do the experimental band gap values correlate with those calculated by the DFT method?
  13. Why did the emission peak position in the range of 550-600 nm shift to the right when adding CuS to the composite MoS2/Bi2WO4 (Figure 4B)? Please explain.
  14. What substance is indicated by the abbreviation AA (line 269)? Please specify.
  15. The paper describes a curve V (line 273), but it is not observed on Figure 6B. Please correct.
  16. Please correct the aptamer concentration values (lines 311-313): μM or μmol/L.
  17. The text mentions Table 2 (line 317), but it is missing.
  18. The text says that for the sensor the detection limit of diclofenac is 0.03 nM. Please indicate this value on the calibration curve (Figure 8B).
  19. The method of measuring the selectivity of the sensor to diclofenac and the results of photoelectrochemical measurements in the coexiting ion supplement are not clear from the text. Please complete it.
  20. The manuscript strongly requires uniformity in material designations. "СН4N2S" is strange for thiourea (do not use brutto-formulae).
  21. There are some misprints in the paper: Base don (line 81); ,, (line 92); PH=7 (line 155); (ii) (line 258); DFC (line 259); difffferent (line 262); was increasd (line 301); PH (Figure 7B); PH (line 314); 
  22. "It is logically deduced from the superior photoelectric properties of the CuS/MoS2/Bi2WO6-5% composites". What does it mean? Logical consideration must be supported by the experimental data.

Author Response

Dear Editor:

Thank you for your suggestive guidelines. According to your suggestions, I have completely revised this manuscript.

The manuscript reports on a photoelectrochemical sensor based on aptamer-inorganic composite. In the paper, the preparation of main components and construction of this aptasensor for detecting diclofenac is claimed. The sensor obtained was studied for the photoelectrochemical response at different working conditions (applied potential, pH and ion supplement). The paper is within the scope of Nanomaterials journal, its results are of certain interest for the readership of the journal. The manuscript is suitable for publication in Nanomaterials journal upon the revision.

  1. In the experimental section, the synthesis conditions (temperature and duration) for preparation of Bi2WO6, MoS2/Bi2WO6 and CuS/MoS2/Bi2WO6 are different (20 or 24 h, 180 or 200C). Please specify why such conditions of synthesis at each stage?

Response: Bi2WO6 was first synthesized under hydrothermal conditions at 180°C for 20h. In order to better introduce MoS2 without generating new phases, 200°C for 24h was screened. Similarly, 200°C for 12h was screened to further compound CuS.

  1. How was the CS/CuS/MoS2/Bi2WO6 suspension obtained? The first step of construction of PEC aptasensor isn't quite clear.

Response: This is our expression error, and we added “Firstly, 20 μL of CuS/MoS2/Bi2WO6 suspension was modified onto ITO within a fixed geometric region (0.5 cm×1 cm) and dried under room temperature to form a CuS/MoS2/Bi2WO6 electrode. Then, 20 μL of CS (0.05%), acting as fixing agent, was dropped onto the CuS/MoS2/Bi2WO6 and dried off at room temperature.” in 2.4 section.

  1. Please specify the chemical composition of anatase polymorph, is it referred to CuS, MoS2 or Bi2WO6? Anatase is TiO2, no titania in the material.

Response: This is our expression error, we have changed anatase polymorph to crystal faces.  

  1. The JCPDS card №39-0256 relates to the orthorhombic phase of Bi2WO6, not tetragonal. The JCPDS card №37-1492 state the hexagonal phase of MoS2, not tetragonal.

Response: We have changed it according for your suggestions.

  1. From the experimental part, it is unclear which component is 5%. Are these mass or molar percentages? The Supplementary also contain the data about 2 and 7% of the component. Why was the composition containing exactly 5% chosen as the main material?

Response: These are mass percentages, and we added “The material is named MoS2/Bi2WO6-2%, MoS2/Bi2WO6-5% and MoS2/Bi2WO6-7% (2%, 5%, 7%= m(MoS2)/[m(MoS2)+m(Bi2WO6)]).”, The impedance spectra and PEC responses of the MoS2/Bi2WO6 electrode were represented in Figure S2, MoS2/Bi2WO6-5% is higher photocurrent and the lowest impedance than for the MoS2/Bi2WO6-2% and MoS2/Bi2WO6-7%, that's why it was chosen to contain exactly 5% as the main material.

  1. The chemical composition of the composite (CS/CuS/MoS2/Bi2WO6) should be established. FTIR spectrum is necessary.

Response: This is our expression error, we have changed "CS/CuS/MoS2/Bi2WO6" to "CuS/MoS2/Bi2WO6" and have been structurally characterized. While CS is drop-coated by the formulated 0.05% CS solution.

  1. In the XPS results, some energy values in the text differ from the energy values on the XPS spectra. Please specify if this is experimental or literature data.

Response: Thanks for your suggestions, this is experimental data.

  1. The paper describes satellite peak for copper at 941.5 eV, but it is not observed on Figure 2E. Please correct.

Response: Thanks for your suggestions, we have deleted it.

  1. The text shows the values of the interplanar spacing for (002) of MoS2, and (110) is highlighted on the HRTEM image (Figure 3D). It should be noticed that the interplanar spacing for (110) plane of MoS2 is 0.158 nm, not 0.615 nm (this is true for the (002) plane), as it shown on Figure 3D. Please mark the plane (110) on the HRTEM image or change the text.

Response: Thanks for your suggestions, we have changed the text.

  1. Please bring the scale labels on the SEM and TEM images to a uniform view.

Response: Thanks for your suggestions, we have changed it.

  1. Please complete the caption to Figure 3: HRTEM image ... (D).

Response: We have added it in the caption to figure 3.

  1. Please estimate the value of the band gap of the obtained materials from the diffuse reflection spectra. How do the experimental band gap values correlate with those calculated by the DFT method?

Response: The estimated band gap value of the resulting material from diffuse reflectance spectroscopy is 2.13 eV in figure S4, the closer the band gap value calculated by the DFT method is to the experimental band gap value, the smaller the error is

  1. Why did the emission peak position in the range of 550-600 nm shift to the right when adding CuS to the composite MoS2/Bi2WO4 (Figure 4B)? Please explain.

Response: We added “the emission peak position in the range of 550-600 nm shift to the right when adding CuS to the composite MoS2/Bi2WO4, which is due to the narrower energy band of the composite material and the red shift of the emission peak occurs [41].”

  1. What substance is indicated by the abbreviation AA (line 269)? Please specify.

Response: The abbreviation AA indicated ascorbic acid, and we have added the ascorbic acid in the manuscript.

  1. The paper describes a curve V (line 273), but it is not observed on Figure 6B. Please correct.

Response: We have changed “curve V” to “curve IV”.

  1. Please correct the aptamer concentration values (lines 311-313): μM or μmol/L.

Response: We have changed it to μmol/L.

  1. The text mentions Table 2 (line 317), but it is missing.

Response: Thanks for your suggestions, we have deleted it.

  1. The text says that for the sensor the detection limit of diclofenac is 0.03 nM. Please indicate this value on the calibration curve (Figure 8B).

Response: Thanks for your suggestions, we have added it in figure 8B.

  1. The method of measuring the selectivity of the sensor to diclofenac and the results of photoelectrochemical measurements in the coexiting ion supplement are not clear from the text. Please complete it.

Response: We added “According to Figure 8C, after waiting for the photocurrent of ap-tamer/CS/CuS/MoS2/Bi2WO6-5% to stabilize, add additional interfering agents include kanamycin (KAN), sulfadimethoxine (SDM), ochratoxin A (OTA), dopamine (DOP), glucose (GLU), Al3+, and Fe2+ respectively, and test the change of photocurrent after stabilization, the photocurrent change showed no discernible change in the coexisting ion supplement.” to the 3.4 section.

  1. The manuscript strongly requires uniformity in material designations. "СН4N2S" is strange for thiourea (do not use brutto-formulae).

Response: Thanks for your suggestions, we have changed it to SC(NH2)2.

  1. There are some misprints in the paper: Base don (line 81); ,, (line 92); PH=7 (line 155); (ii) (line 258); DFC (line 259); difffferent (line 262); was increasd (line 301); PH (Figure 7B); PH (line 314); 

Response: We have changed it according for your suggestions.

  1. "It is logically deduced from the superior photoelectric properties of the CuS/MoS2/Bi2WO6-5% composites". What does it mean? Logical consideration must be supported by the experimental data.

Response: This is our expression error, we have deleted it.

Reviewer 2 Report

Comments: This article reports a visible-light-active CuS/MoS2/Bi2WO6 aptamer sensitively detects the non-steroidal anti-inflammatory drug diclofenac. The structure of the synthesized materials has been characterized well, and these analyses are reasonable. However, authors should address the following comments for its acceptance.

1.    Author needs to index the peaks in the XRD patterns of CuS, Bi2WO6 MoS2/Bi2WO6-5%, and CuS/MoS2/Bi2WO6-5% composites for a clear understanding of readers.

2.    Author should supply the XPS survey spectrum of CuS/MoS2/Bi2WO6-5% composites to determine their elemental composition and states. Besides, the O 1s level needs to be deconvoluted for the indemnification of functionalities.

3.    What about the EDX spectrum of the prepared CuS/MoS2/Bi2WO6-5%.

4.    EIS Nyquist plots fitting curves of CuS/MoS2/Bi2WO6-5%, (II) CS/CuS/MoS2/Bi2WO6-5%, (III) aptamer/CS/CuS/MoS2/Bi2WO6-5% and (IV) DCF/aptamer/CuS/MoS2/Bi2WO6-5% are missing. Hence, author should be fitted to determine the electrochemical performance between electrode and eletrolytres.

5.    What about the reusability of the prepared CuS/MoS2/Bi2WO6 towards sensing the drug.

6.    Conclusion needs to be improved with novelty and the importance of the present study.

Author Response

Dear Editor:

Thank you for your suggestive guidelines. According to your suggestions, I have completely revised this manuscript.

Comments: This article reports a visible-light-active CuS/MoS2/Bi2WO6 aptamer sensitively detects the non-steroidal anti-inflammatory drug diclofenac. The structure of the synthesized materials has been characterized well, and these analyses are reasonable. However, authors should address the following comments for its acceptance.

  1. Author needs to index the peaks in the XRD patterns of CuS,Bi2WO6 MoS2/Bi2WO6-5%, and CuS/MoS2/Bi2WO6-5% composites for a clear understanding of readers.

Response: We have indexed the peaks according for your suggestions.

  1. Author should supply the XPS survey spectrum of CuS/MoS2/Bi2WO6-5% composites to determine their elemental composition and states. Besides, the O 1s level needs to be deconvoluted for the indemnification of functionalities.

Response: We have deconvolved the O 1s level to deconvolute for the indemnification of functionalities.

  1. What about the EDX spectrum of the prepared CuS/MoS2/Bi2WO6-5%.

Response: We have added the EDX spectrum of the prepared CuS/MoS2/Bi2WO6-5% in Figure S3.

  1. EIS Nyquist plots fitting curves of CuS/MoS2/Bi2WO6-5%, (II) CS/CuS/MoS2/Bi2WO6-5%, (III) aptamer/CS/CuS/MoS2/Bi2WO6-5% and (IV) DCF/aptamer/CuS/MoS2/Bi2WO6-5% are missing. Hence, author should be fitted to determine the electrochemical performance between electrode and eletrolytres.

Response: Thanks for your suggestions, we have added the EIS Nyquist plots fitting curves in figure 6.

  1. What about the reusability of the prepared CuS/MoS2/Bi2WO6 towards sensing the drug.

Response: We use the same electrode for sensing the drug in table 1, and the current density remains stable, indicating that the material has good reusability. 6.    Conclusion needs to be improved with novelty and the importance of the present study.

Response: We have improved with novelty and the importance of the present study in conclusion.

Round 2

Reviewer 1 Report

The authors corrected most of the items, but not all of my comments were met.   The authors' reply to point №4 (the description of XRD results) did not meet the important concern: The JCPDS card №37-1492 is for the hexagonal MoS2, not orthorombic. This point is not reflected in the text.   The authors changed "CS/CuS/MoS2/Bi2WO6" to "CuS/MoS2/Bi2WO6" according to point №6. FTIR data was not provided for any of these composites. Using of thiourea for the synthesis of CuS/MoS2/Bi2WO6 could result in organic residuals and affect the photoelectrochemical properties of the material, including the detection efficiency.   In the XPS results, some energy values in the text still differ from the energy values in the XPS spectra. The authors claimed that this is experimental data, but the text and figures (2A, 2E, 2F) again do not correspond to each other. What are the reasons for such differences in energy values?   The authors reply to point №9 that they have changed the text, but the values of the interplanar spacing for (110) of MoS2 haven't been changed (0.615 nm vs 0.158 nm).   The authors clarified the meaning of the abbreviation AA, but the significance of ascorbic acid in electrochemical measurements of the sensor for the detection of diclofenac is still unclear.   The authors added the diclofenac detection limit in Figure 8B, but I meant to visualize calibration data in a wider range (from log(0.03) to log(500)).   Summarizing, the manuscript hasn’t been improved in some important points. Hence, the manuscript still need some more corrections.

Author Response

Dear Editor:

Thank you for your suggestive guidelines. According to your suggestions, I have completely revised this manuscript.

The authors corrected most of the items, but not all of my comments were met.  

The authors' reply to point №4 (the description of XRD results) did not meet the important concern: The JCPDS card №37-1492 is for the hexagonal MoS2, not orthorombic. This point is not reflected in the text.

Response: Thanks for your suggestions, we have checked and changed it.

The authors changed "CS/CuS/MoS2/Bi2WO6" to "CuS/MoS2/Bi2WO6" according to point №6. FTIR data was not provided for any of these composites. Using of thiourea for the synthesis of CuS/MoS2/Bi2WO6 could result in organic residuals and affect the photoelectrochemical properties of the material, including the detection efficiency.

Response: We added the FTIR in Figure S4,and added “Figure S4 shows the FTIR spectra of the as-prepared samples. Specifically, the adsorption band at 442.3 cm−1  is owing to the bending vibrations of Bi–O bond. A vibration peak at 596.5 cm−1 signifies the presence of Cu-S bond. the strong peaks at 778.1 cm−1 and 823.8 cm−1 are attributed to the W-O and W-O-W bond of Bi2WO6, respectively. The bands at 1380 and 3430 cm−1 are induced by the bending and stretching vibrations of the water molecules adsorbed on the sample surface. The results show that there is no residue of other organic matter” in the supporting information.

In the XPS results, some energy values in the text still differ from the energy values in the XPS spectra. The authors claimed that this is experimental data, but the text and figures (2A, 2E, 2F) again do not correspond to each other. What are the reasons for such differences in energy values?

Response: Thanks for your suggestions, we have checked and changed it in the XPS results, make some energy values in the text match those in the XPS spectrum.

The authors reply to point №9 that they have changed the text, but the values of the interplanar spacing for (110) of MoS2 haven't been changed (0.615 nm vs 0.158 nm).

Response: Sorry for our mistakes, we have changed the values of interplanar spacing for (110) on the HRTEM image and the text.

The authors clarified the meaning of the abbreviation AA, but the significance of ascorbic acid in electrochemical measurements of the sensor for the detection of diclofenac is still unclear.

Response: We added “Ascorbic acid as an electron donor can consume holes in semiconductor materials, reduce the electron-hole recombination rate of materials, and amplify the photocurrent signal.” in the 3.3 section.

The authors added the diclofenac detection limit in Figure 8B, but I meant to visualize calibration data in a wider range (from log(0.03) to log(500)).   Summarizing, the manuscript hasn’t been improved in some important points. Hence, the manuscript still need some more corrections.

Response: Thanks for your suggestions, We assayed and screened diclofenac for a larger range of concentrations and showed good linearity over the 0.03-500 nM range. In future work, we will also further improve the material to achieve a wider detection range.

Reviewer 2 Report

The authors have moderately addressed the issues raised by the reviewers. Hence, the revised version of the manuscript may acceptable to the journal standard.

Author Response

Dear Editor:

Thank you for your suggestive guidelines. According to your suggestions, I have completely revised this manuscript.

The authors have moderately addressed the issues raised by the reviewers. Hence, the revised version of the manuscript may acceptable to the journal standard.

Response: Thanks for your suggestions, good luck with your work.
